# CA Modeling of Microsegregation and Growth of Equiaxed Dendrites in the Binary Al-Mg Alloy

**DOI:** 10.3390/ma14123393

**Published:** 2021-06-18

**Authors:** Andrzej Zyska

**Affiliations:** Faculty of Production Engineering and Materials Technology, Czestochowa University of Technology, 19 Armii Krajowej Av, 42-200 Czestochowa, Poland; andrzej.zyska@pcz.pl

**Keywords:** CA model, Al-Mg alloy, dendritic solidification, microsegregation

## Abstract

A two-dimensional model based on the Cellular Automaton (CA) technique for simulating free dendritic growth in the binary Al + 5 wt.% alloy was presented. In the model, the local increment of the solid fraction was calculated using a methodology that takes into account changes in the concentration of the liquid and solid phase component in the interface cells during the solidification transition. The procedure of discarding the alloy component to the cells in the immediate vicinity was used to describe the initial, unstable dendrite growth phase under transient diffusion conditions. Numerical simulations of solidification were performed for a single dendrite using cooling rates of 5 K/s, 25 K/s and 45 K/s and for many crystals assuming the boundary condition of the third kind (Newton). The formation and growth of primary and secondary branches as well as the development of component microsegregation in the liquid and solid phase during solidification of the investigated alloy were analysed. It was found that with an increase in the cooling rate, the dendrite morphology changes, its cross-section and the distance between the secondary arms decrease, while the degree of component microsegregation and temperature recalescence in the initial stage of solidification increase. In order to determine the potential of the numerical model, the simulation results were compared with the predictions of the Lipton-Glicksman-Kurz (LGK) analytical model and the experimental solidification tests. It was demonstrated that the variability of the dendrite tip diameter and the growth rate determined in the Cellular Automaton (CA) model are similar to the values obtained in the LGK model. As part of the solidification tests carried out using the Derivative Differential Thermal Analysis (DDTA) method, a good fit of the CA model was established in terms of the shape of the solidification curves as well as the location of the characteristic phase transition temperatures and transformation time. Comparative tests of the real structure of the Al + 5 wt.% Mg alloy with the simulated structure were also carried out, and the compliance of the Secondary Dendrite Arm Spacing (SDAS) parameter and magnesium concentration profiles on the cross-section of the secondary dendrites arms was assessed.

## 1. Introduction

Dendritic growth is one of the fundamental problems present in metallurgical research. The understanding and mathematical description of physical phenomena occurring during dendrite growth play a fundamental role in predicting the structure and properties of metals and alloys. The solidification process can be analysed on the basis of experimental tests, analytical models, and numerical models [1,2,3,4,5]. The advantage of numerical models is the possibility to dynamically track the movement of the interface and generate images of modelled structures.

Structure modelling in castings concerns a single crystal or a group of crystals, and is carried out for a selected fragment of a casting or for its entire area [6,7,8,9,10,11]. In most cases, the growth of equiaxial crystals formed under the conditions of relatively uniform heat dissipation in all directions, that correspond to volume crystallisation, is modelled. This type of crystals is usually dominant in the mass of castings produced by basic casting technologies. In some solutions, the growth of columnar crystals that arise under directional heat dissipation is modelled and the transition zone from columnar to equiaxial crystals is defined. (CET—Columnar to Equiaxed Transition).

Currently, the phase field (PF) method and the cellular automaton (CA) method are most frequently used to simulate the numerical structure. Models based on the CA technique are characterised by a simple structure, clear interpretation, and the possibility of direct implementation of mathematical equations describing a number of simultaneous physical phenomena [12,13,14]. 

CA models are mainly used to model dendrite morphology. The modelling is based on the fundamental principles of the crystallisation theory of metals and alloys. The reproduction of the shapes of growing dendrites occurs as a result of tracing the movement of the solidification front in the automaton cells with a size of 1μm. Morphological models contain mathematical formulas describing the liquid/solid interface: its energy, curvature, anisotropy, and undercooling. At the microscale level, the diffusion equation and the heat conduction equation, and sometimes also the equation of motion of the liquid metal, are solved. Moreover, any crystal nucleation mechanism can be adapted to the cellular automaton. Including all these phenomena in one model allows for recreating the actual solidification process to a large extent. Numerical simulations make it possible to trace the formation and development of first and higher order arms in dendrites or the coupled phase growth during eutectic solidification, taking the interaction of growing crystallites into account. Information on the topology of the modelled structures and on the microsegregation of alloy components in the liquid and solid phases is obtained [3,8,9,10,11,12,13,14,15,16,17,18,19,20,21,22].

Growth rate is the basic physical quantity that describes the movement of the crystallisation front and the evolution of the structure in numerical morphological models. This rate is determined on the basis of the Stefan condition, the lever rule, concentration and kinetic supercooling [23,24,25,26,27,28,29]. Modelling of the dynamics of the solidification front at the microscale level using the CA technique was first presented by Dilthey and Pawlik [23]. In the calculation algorithm, the authors included the mass and energy transport equations as well as the balance of components on the mobile liquid-solid interface. In the following years, Beltran-Sanchez and Stefanescu [15] developed a model based on the concept of dendrite growth under steady-state conditions. In this approach, the movement of the interface is controlled by the diffusion rate of the component rejected before the solidification front. The model uses a proprietary methodology to solve the diffusion equation in interface cells, as well as a modified solid fraction growth equation with a term that causes small random concentration fluctuations. The numerical model was validated in terms of quantity and quality. The results were found to be in fair agreement between the simulated distance between the secondary dendrite arms (SDAS) and the literature data, and between the simulated values of the dendrite front movement rate and predictions of the Lipton-Glicksman-Kurz (LGK) solution for various concentration undercooling of the alloy.

Zhu and Stefanescu [18] developed a model in which the velocity of the liquid-solid interface depends on the difference between the instantaneous local equilibrium concentration and the instantaneous local actual concentration of the component in the liquid phase at the solidification front. In the model, this difference was determined on the basis of the local temperature and interface curvature, and the solution of the diffusion equation. The proposed methodology for calculating the solid fraction increment in interphase cells was successfully applied in other studies [19,24,25,26]. It is possible to simulate the evolution of single and multiple dendrites in 2D and 3D space, as well as the formation and development of spherical grains in liquid-solid processes by means of cellular automata. The images of microstructures obtained with all CA models are very realistic and qualitatively consistent with the results of experimental tests.

One of the problems in developing numerical CA solidification models is the correct calculation of the instantaneous composition of the liquid and solid phases in the interface cells. The finite dimension of the cells and approximate numerical methods used to solve the diffusion equation may lead to obtaining excessive values of the alloy component concentration in the interface cell with an increase in the solid fraction. This problem was found by several researchers [15,17,18,27]. To reduce overproduction of a component, interface cells are treated as liquid phase cells when solving the mass transport equation. Moreover, in most diffusion-controlled CA models [15,17,18,24,25,26], the growth rate is calculated assuming a local equilibrium on the liquid-solid interface. This assumption can only be considered valid in the case of steady-state growth. In the initial period of unstable growth, the amount of the rejected component is too high to be transported from the solidification front by a diffusion mechanism. Therefore, the acceptance of the mass balance condition at the solidification front cannot be satisfactory for this period [18].

The article presents a 2D model for the numerical simulation of dendritic solidification of a two-component alloy on the example of Al + 5 wt.% Mg. While building the CA model, particular attention was paid to the problem of calculating the alloy component concentration in the interface cells. The growth rule was based on the mass balance of the component to calculate the appropriate component concentrations at the solidification front and in the liquid and solid phases. At the same time, limitations on the maximum speed of solid phase growth in the time interval were introduced. For the adopted growth rule, the diffusion equation was solved in two stages with a pseudo-initial condition. The developed methodology made it possible to determine the instantaneous component concentrations at the solidification front, taking concentration and capillary undercooling into account. For the initial period of unstable growth, the procedure of discarding component to the nearest neighbouring cells was applied. The multi-grid method was also used in the paper to solve the heat transport equation in order to shorten the calculation time. The developed model allows for a realistic reconstruction of the dendrite morphology during solidification of the binary alloy and is consistent with the results of the Derivative Differential Thermal Analysis (DDTA) experiment and analyses of the chemical composition of EDS.

## 2. Description of the Model

Modelling of the dendritic structure evolution was performed using the cellular automaton technique in combination with the control volume method. The calculations were performed on a flat area divided by a regular square grid into elementary cells with side *a*. The following basic quantities were defined on the automaton grid: temperature—*T*, concentration of the ingredient—*W*, solid phase fraction—*f*_S_, preferential angle of crystallographic orientation—θ_0_, interface curvature—*K*, direction normal to the solidification front—φ. During solidification, each cell changed its phase state from liquid (*f*_S_ = 0) to transitional (0 < *f*_S_ < 1) and finally to solid state (*f*_S_ = 1). Initially, the temperature and concentration distribution across the domain was homogeneous and consistent with the values *T*_0_ and *C*_0_. The model uses the von Neumann neighbourhood to solve the mass and energy transport equations, and the Moor neighbourhood to determine the remaining quantities. In order to reduce the artificial anisotropy of dendritic growth induced by the CA grid, the procedure of following the preferred growth directions was applied [30]. The quantitative and qualitative description of the growth kinetics of dendritic crystals is represented by three basic fields: solid fraction field, concentration field and temperature field. The instantaneous interface location and the instantaneous shape of dendritic structures are recreated by conjugate solving of the system of model equations.

The concept of the model assumes that the dendrite growth rate is controlled by the component discharge rate from the solidification front by the diffusion mechanism. The growth driving force is determined by the deviation from the equilibrium state, and its measure is determined by the difference between the instantaneous local equilibrium concentration and the instantaneous local chemical composition of the liquid phase. The equilibrium concentration results from the thermodynamic relationship between the liquidus temperature and the front curvature. The instantaneous temperature field and the front shape determine the temporary distribution of the equilibrium concentration in the interface cells, while the concentration gradient formed in the front determines the growth rate of the solid phase.

### 2.1. Modelling of Component Diffusion and Solid Phase Growth

The 2D diffusion equation with the effective diffusion coefficient was used to describe the component transport:(1)$$\frac{\partial W(x,y,t)}{\partial t}=D_{ef}\left(\frac{\partial^{2}W(x,y,t)}{\partial x^{2}}+\frac{\partial^{2}W(x,y,t)}{\partial y^{2}}\right)$$

Effective Diffusion Coefficient *D_ef_* depends on the phase state of the central cell and the configuration of the cells of the immediate surroundings at the moment *t* and takes homo- and heterogeneous neighbourhoods into account:(2)$$D_{ef}=\left\{ \begin{array}{ll} 0.5\;(D_{S}+D_{L}) & for\;\;(0<f_{S,i,j}<1)\;\;\;and\;\;\;(0<f_{S,Ne}<1)\\ k_{0}D_{S} & for\;\;(f_{S,i,j}>0)\;\;\;and\;\;\;(f_{S,Ne}=1)\\ D_{L} & for\;\;(f_{S,i,j}<1)\;\;\;and\;\;\;(f_{S,Ne}=0) \end{array}\right.$$
where: *f_S,i,j_, f_S,Ne_* are solid phase fractions in the central cells *i,j* and in neighbouring cells (von Neumann neighbourhood), respectively, *D_S_, D_L_* are the diffusion coefficients of the component in the liquid and solid phases, respectively, and *k*_0_ is the component separation coefficient.

During the growth of dendritic structures, interphase boundaries can be concave, flat and convex, and the surface tension demonstrates anisotropy in the privileged crystallographic directions (of crystal growth). As a result, there are local changes in the equilibrium solidification temperature at the front at the same component concentration. The paper uses the modified Gibbs-Thomson equation, which allows for determining the change in the equilibrium temperature in the function of concentration undercooling and the solidification front curvature:(3)$$T^{F}=T_{L}+m_{L}(W_{L}^{F}-W_{0})-\Gamma \{1-\delta \cos\left[m_{S}(\varphi -\theta_{0})\right]\}K$$
where: *θ*_0_—preferential angle of crystallographic orientation, *δ*—surface tension amplitude, *m_s_*—crystal symmetry coefficient (*m_s_* = 4), *K*—interface curvature, φ—direction normal to the solidification front, *m*_L_—direction coefficient of the liquidus line, G—Gibbs-Thomson coefficient, *T^F^*—solidification front temperature, *T_L_*—liquidus temperature, *W*_0_—the initial component concentration.

By transforming Equation (3), the dependence that determines the equilibrium component concentration on the interface is obtained:(4)$$W_{L}^{F}=W_{0}+\frac{1}{m_{L}}\left(T^{F}-T_{L}+\Gamma K\left\{1-\delta \cos[m_{S}(\varphi -\theta_{0})]\right\}\right)$$

To calculate the instantaneous increments in the solid fraction in the interface cells, the mass balance equation, which after differentiation with respect to time and approximation by the differential quotient is expressed by the following formula, was used:(5)$$\begin{array}{cc} k_{0}W_{L,i,j}^{t}\Delta f_{S,i,j}+k_{0}(W_{L,i,j}^{t+1}-W_{L,i,j}^{t})f_{S,i,j}^{t}-W_{L}^{t}\Delta f_{S,i,j}+ & \\ (W_{L,i,j}^{t+1}-W_{L,i,j}^{t})(1-f_{S,i,j}^{t})= & \Delta W_{i,j} \end{array}$$

Calculations are performed in two stages. In the first stage, mass transport is also solved and the change in concentration ∆*W* is determined assuming that the increment of the solid fraction is zero (Δ*f_S_* = 0, pseudo-initial condition). As a result, the component concentration in the cells of the interface decreases to the value:(6)$$W_{L,i,j}^{t+1,pseudo}=W_{L,i,j}^{t}+\frac{\Delta W_{i,j}}{1+f_{S,i,j}^{t}(k_{0}-1)}$$

In the second stage, it is checked which cells of the concentration interface $W_{L,i,j}^{t+1,pseudo}$ are less than the equilibrium concentration $W_{L,i,j}^{F}$. For cells in which the condition is met $W_{L,i,j}^{t+1,pseudo}<W_{L,i,j}^{F}$ the value of the equilibrium concentration is entered $W_{L,i,j}^{t+1}=W_{L,i,j}^{F}$ and it determines the increment of the solid phase on the basis of the transformed Formula (5):(7)$$\Delta f_{S,i,j}^{}=\frac{\Delta W_{i,j}-(W_{L,i,j}^{F}-W_{L,i,j}^{t})[1+f_{S,i,j}^{t}(k_{0}-1)]}{W_{L,i,j}^{t}(k_{0}-1)}$$

In the remaining cells of the interface, the concentration values are determined from the pseudo-initial condition $W_{L,i,j}^{t+1}=W_{L,i,j}^{t+1,pseudo}$. The chemical composition of liquid and solid cells is also determined using Equation (6) substituting, accordingly, *f_S_* = 0 and *f_S_* = 1.

Based on the calculated increments of the solid fraction for each step Δ*t ^f^* the normal growth rate υ_n_ is determined from dependence [15,18,19]:(8)$$\upsilon_{n}=\Delta f_{s}\frac{\Delta x}{\Delta t^{f}}$$

In the adopted procedure, a temporary “stop” of the solidification front allows for determining the instantaneous component discharge rate from the interface cells and its concentration in the liquid phase. Then, by comparing the value of this concentration with the value of the equilibrium concentration, it is possible to determine the cells that are involved in the transformation and their deviation from the equilibrium state. The size of the increments of the solid phase is essentially influenced by the concentration gradient on the side of the liquid phase, as well as the shape of the front curvature and the temperature field, whereby the last two variables are used to determine the local equilibrium concentration. Moreover, the proposed method of calculating the increments of the solid fraction takes into account the change in the component concentration in the solid and liquid phases during the passage of the solidification front through the interface cell.

Solving Equation (4) requires determining the direction normal to the solidification front in the interface cell. Three methods of determining this value are most often used in CA models. The first one uses the procedure of tracking the position and topology of the interface [31,32], the second one is based on the determination of the centre of mass of the block of cells in the closest surrounding [27,30], and in the third one, calculations are performed on the basis of solid fraction gradients. In order to ensure quick calculations while maintaining high accuracy of the results, the last of the following variants was applied to the model:(9)$$\varphi =\arctan\left[\left(\frac{\partial f_{S}}{\partial x}\right){\left(\frac{\partial f_{S}}{\partial y}\right)}^{-1}\right]$$

The last variable in Equation (4) is the interface curvature. Also, in this case, several methods of determining the shape of the solidification front can be used [15,18,19,33]. The quickest, but the least precise, are the procedures for counting the solids fraction on cell blocks 3 × 3 or 5 × 5. The methods that use solid fraction gradients are more accurate. The modelling uses the dependence that was first proposed by Kothe [34]: the interface curvature is determined on the basis of first and second order partial derivatives:(10)K=[2∂fS∂x∂fS∂y∂2fS∂x∂y−(∂fS∂x)2∂2fS∂y2−(∂fS∂y)2∂2fS∂x2]×[(∂fS∂x)2+(∂fS∂y)2] −32

### 2.2. Temperature Field Approximation and Interpolation

Solving a thermal problem on a cellular automaton grid with the use of explicit diagrams is numerically ineffective. The stability of these methods [18,26] requires limiting the time step to approximately 10^−9^ s in the case of modelling the Al alloys solidification on a CA grid with a dimension of 1 µm. In order to overcome this inconvenience, implicit methods are used, or a temperature that varies in time, but is homogeneous, is assumed for the entire calculating area [4,12,13,14]. Another variant that was adapted to the model is the determination of the temperature field using two grids. A sparse grid was superimposed on the main CA grid, the constant of which (Δ*x_B_*) is several times larger than the automaton cell dimension *a*. A block of cells with dimension *n_B_* × *n_B_* on the main grid corresponds to a single cell of the sparse grid. Feedback occurs between the grids. On the sparse grid, for each time interval, the heat transport equation is solved, and the determined temperature field is interpolated into the main grid of the automaton. In turn, the instantaneous increments of the solid fraction calculated in CA unit cells are summed up on blocks and transferred to the sparse grid.

The temperature field in the domain discretized by the sparse grid was calculated on the basis of the equation:(11)$$\frac{\partial T(x,y,t)}{\partial t}={\bar{a}}_{B}\left(\frac{\partial^{2}T(x,y,t)}{\partial x^{2}}+\frac{\partial^{2}T(x,y,t)}{\partial y^{2}}\right)+\frac{L}{c}\frac{\partial f_{SB}}{\partial t}$$
where: *c*—specific heat, *L*—heat of solidification, ${\bar{a}}_{B}$—average thermal diffusivity.

The increments of the solid fraction in the sparse grid cells were calculated from the dependence:(12)$$\Delta f_{SB}=\frac{\sum \Delta f_{S}}{n_{B}^{}\times n_{B}^{}}$$

The interpolation of the temperature field onto the main grid of the automaton was carried out based on the procedure of a systematic transition from the distant to the nearest neighbourhood (Figure 1). The calculations start with solving the heat transport equation and determining the temperature field on the sparse grid (Figure 1a). Then the obtained temperatures are transferred to the central cells of the CA main grid (Figure 1b). The blocks on the main grid were placed in such a way so that the dimension of a single block *n*_B_ × *n*_B_ met the relation 2*^n^* × 2*^n^* elementary cells (*n*—natural number). The temperature interpolation from the central cells to the entire main grid is carried out in 2*n* steps. In the first step, the temperatures in the cells $\left(\pm 2^{n-1}a\sqrt{2^{}},\pm 2^{n-1}a\sqrt{2^{}}\right)$ located from central cells are calculated. The obtained values together with the rewritten ones constitute the input data for the second step, in which the temperatures in the cells from the environment are determined $\left(\pm 2^{n-1}a,\pm 2^{n-1}a\right)$. In the third and fourth steps, interpolation is performed based on the neighbourhoods $\left(\pm 2^{n-2}a\sqrt{2^{}},\pm 2^{n-2}a\sqrt{2^{}}\right)$ and $\left(\pm 2^{n-2}a,\pm 2^{n-2}a\right)$. The procedure is continued by systematically reducing the distance between cells in the main and intermediate directions by half (according to the wind rose). Moving from the distant to the nearest neighbourhood, the temperatures in all cells of the automaton are determined. Since the distance between the cells is the same in each step, the interpolation problem comes down to calculating the mean of the four surrounding cells. The interpolation scheme on a fragment of the main grid with blocks with a dimension of 2^3^ × 2^3^ for the next two steps is illustrated in Figure 1c,d. The diagram shows the solution using the periodic boundary condition at the ends of the cellular automaton. The sparse grid size is determined by the ratio Δx_B_ = 2^n^a.

### 2.3. Nucleation and Initial Growth Period under Transient Diffusion Conditions

The cellular automaton technique allows for modelling of the nucleation stage using deterministic or probabilistic relationships. The models use the variants of immediate and continuous nucleation. The initial distribution of cells called nuclei in the calculating domain is usually determined at random, after reaching the appropriate temperature (depending on the adopted nucleation model). Neighbouring cells are joined to *nucleus cells* to form an interface that evolves based on accepted growth rules. In the presented model, it was assumed that nucleation occurs immediately, and that the solidification process begins after reaching the nucleation temperature *T_N_* which is less than *T_L_*. Temperature *T_N_* depends directly on the rate of cooling and diffusion of the component in the liquid phase. The idea of the algorithm is as follows. Once the liquidus temperature is reached, a randomised location of *nucleus cells* is determined in the CA area, and it is assumed that the proportion of the solid phase in them is one. The formation of *nucleus cells* is related to the “production” of an excess of the alloy component Δ*W = W_L_ − W_S_*. It is assumed that the component is evenly rejected to all cells in the Moore vicinity and its concentration in each of these cells is increased by the amount Δ*W/8*. From that moment on, the procedure for solving the diffusion equation is started. The initial increase in the component concentration around *nucleus cells* makes them inactive for a certain period of time. Only as a result of channelling the component deep into the bath and dissipating superheat, they are able to connect their neighbours. This occurs as soon as neighbouring cells reach temperature *T^F^*, determined by Equation (3). Temperature *T^F^* is considered the nucleation temperature *T_N,_* and the time needed for lowering the temperature from *T_L_* to *T_N_* an incubation period.

### 2.4. Optimisation of the Time Step for Model Equations

The use of the iterative process for explicit numerical schemes described in the previous chapters is related to the determination of the limit value of time step ∆*t_stab_* which ensures the stability of the calculations. The stability condition for the numerical solution of the diffusion equation is expressed by the dependence:(13)$$\frac{D_{L}\Delta t^{W}}{\Delta x^{2}}<0.25$$
while the energy equations on the sparse grid ∆*x*_B_:(14)$$\frac{a_{B}\Delta t^{T}}{\Delta x_{B}^{2}}<0.25$$

Apart from these two components, step ∆*t_stab_* is also limited by an additional criterion that relates to the speed at which the solidification front moves through the interface cell. In order to ensure high “precision” of calculations, the increment of the solid fraction in one step of the calculations must be sufficiently small. The construction of the automaton assumes that the cell changes its state from transient to solidified at the moment when the solid fraction reaches one, and exceeding this value is omitted. Moreover, due to the use of an explicit scheme to solve the diffusion equation, excessive solid phase growth can lead to overestimated values of the component concentration in the interface cells. It was established in [27] that the elementary increment of the solid phase ∆*f*_el_ in one-time step should be in the range (0.01–0.1). Following these guidelines and preliminary simulations, it was assumed that the maximum increment in one step of the calculations was 0.02. By inserting this value into Equation (8) and taking dependencies (13) and (14) into account, the time step limitation is obtained in the form of the formula:(15)$$\Delta t_{stab}=\min\left(0.25\frac{\Delta x^{2}}{D_{L}},0.02\frac{\Delta x}{\upsilon_{n}^{\max}},0.25\frac{\Delta x_{B}^{2}}{a_{B}}\right)$$

Above the liquidus temperature, only the thermal problem for which ∆*t_stab_ =* ∆*t^T^* is solved. After the liquidus temperature is exceeded, optimization of the time step is done with regard to quantity ∆*t^f^*, comparing it with ∆*t^T^* and ∆*t^W^*. For high solids growth rates, upper limit ∆*t_stab_* is related to the second term of formula (15), while for medium and low velocities, it is related to the stability of the explicit scheme of the thermal conductivity equation. Due to step ∆*t^T^*, despite the use of the multi-grid technique, it can be several dozen times smaller than steps ∆*t^f^* or ∆*t^W^*; in order to shorten the calculating time, the algorithm uses a separate iteration procedure for the thermal problem. In this procedure, the number of iterations based on the ratio ∆*t^f^*/∆*t^T^* or ∆*t^W^/*∆*t^T^* are determined. The isolation of the energy equation means that the remaining model equations can be solved with the limitation for ∆*t^f^* or ∆*t^W^*. Optimization is performed for each iteration, and a set new value ∆*t_stab_* is valid for the next step. No calculations are updated for the current step.

## 3. Numerical Simulations Results

### 3.1. Free Growth of a Single Dendrite

Computer simulations of the free dendrite growth were carried out on the Al + 5 wt.% Mg using the material parameters from references [27,33,35]. The calculations were performed on a cellular automaton with elementary cells with dimension of 160 × 160, which corresponds to a flat area of 320 × 320 µm^2^. The side length of the cell was selected empirically. If the value “*a*” of the cells is selected too high, the shape of the growing main arms of the dendrite ceases to depend on their preferential crystallographic orientation and begins to depend only on the symmetry of the CA grid. For the cooling rates used in the simulations, the growth of dendrites in the direction corresponding to their assumed orientation is obtained for a cell side length of up to 2 µm. The diffusion equation was solved assuming a periodic boundary condition at all ends of the CA grid. For the energy equation, it was assumed that the heat dissipation from the modelled area occurs perpendicular to its surface and at a constant value of the heat flux—the boundary condition of the second type. After exceeding the temperature *T_L_* one *nucleus cell* was placed in the centre of the calculating domain. This cell was assigned an orientation angle that is inconsistent with both of the major directions of the CA grid. Preferential angle of crystallographic orientation θ_0_ was 25°.

The next Figure 2a–c present the results of the simulation of the influence of the cooling rate on the development of dendritic structures and the accompanying instantaneous fields of the component concentration in the liquid, solid and transition phases. The interface cells show the average Mg concentration values resulting from the mass balance. The calculations were made for three cooling rates, the values of which, in the absence of an internal heat source, are: 5 K/s, 25 K/s and 45 K/s.

On the basis of the obtained simulation results, the sequences of free growth of the dendritic crystal can be traced. In its initial stage, the solid phase grows on a square *nucleus cell*, changing the original shape to a circle. This circular form, characteristic of the onset of dendritic solidification, is further disturbed fourfold in mutually perpendicular directions. As a result, the isotropic structure is transformed into a square whose vertices are oriented by the preferential angle. After some time, the growth of the vertices of the square leads to the formation of four first-order branches. For a low cooling rate (5 K/s), only main dendrite branches are formed, while for medium and high cooling rates, characteristic disturbances of the surface of the first-order branches appear, from which the secondary branches crystallize.

As the cooling rate increases, the distance between the axes of the side branches and the transverse dimension of the main branches decrease. At medium and high cooling rates, the first order branches increase their transverse dimensions only in the vicinity of the dendrite front. Secondary branches grow below this zone and inhibit further transverse growth of the main branch. As a result, the main branches have a uniform cross-section along their entire length. Such a property does not occur for the cooling rate of 5 K/s and the largest transverse dimension of the main branch appears above half its length, counting from the base of the dendrite. At the highest cooling rate (45 K/s), the growth rate of the side branches’ front clearly increases. As a result, the difference between the length of the main branches and the length of the side branches decreases.

In models using the CA technique, the equations of thermal conductivity and diffusion are solved in a conjugate manner, which makes it possible to track the change in component concentration in the growing dendrite, at the solidification front and the surrounding liquid phase. On a local scale, a change in the state of the system, which is related to the segregation of the component in the liquid and solid phase, is observed. The degree of component segregation depends mainly on the shape of the formed structures, their characteristic dimensions, the size of the solidification temperature interval, and the diffusion and cooling rate. In the initial solidification period, first of all, the uneven distribution of the alloy component in the liquid phase is observed, while in the solid phase, segregation is negligible due to the very small dimensions of the dendrite and short diffusion paths. Based on Figure 2, it can be concluded that with the free growth of dendrites, the phenomenon of segregation in the liquid phase intensifies with the increase in cooling rate. At a cooling rate of 45 K/s, the liquid phase enriched with the alloy component occupies only a small area around the dendrite arms: the diffusion layer is practically in the immediate vicinity of the solid phase. In the remaining liquid, the component concentration is equal to the initial concentration C_0_.

Figure 3b shows the instantaneous alloy component concentration fields in the successive phases of dendrite growth at a cooling rate of 45 K/s. The profiles were marked along the line being the symmetry axis of the main branch of the dendrite (Figure 3a). From the graph presented in Figure 3b, the instantaneous width of the diffusion layer in front of the dendrite as well as the instantaneous Mg concentrations in the tip interface cells can be read. In the initial stage of solidification, the component concentration at the front increases rapidly and reaches a maximum value of approx. 6.3% Mg. After this period, the crystal begins to grow under the set diffusion conditions and the Mg concentration remains practically at the same level. The highest concentration of the alloy component, regardless of the cooling rate, occurs at the base of the dendrite, in interdendritic spaces (for cooling rates of 25 K/s and 45 K/s) and in front of the main branches’ front. A typical distribution of the alloy component along the section starting from the base of the dendrite and crossing the secondary arms (line 2 in Figure 3a) is presented in graph 3c. The high value of Mg concentration in the distinguished places inhibits the lateral growth of the main branches and contributes to the development of secondary branches, the fronts of which move towards the zone with a lower component concentration.

Lowering the cooling rate to 25 K/s causes the zone of the liquid phase enriched with alloy component to increase and the maximum Mg concentration values in the interface cells to decrease. Lower cooling rates correspond to lower concentration gradients in front of the solidification front and, as a result, lower crystal growth rates and larger distances between secondary branches. For a low cooling rate (5 K/s), the alloy component segregation phenomenon is not so intensely visible. The area of the enriched liquid phase becomes more blurred, and the Mg concentration in the transition cells at the beginning of solidification does not exceed 5.4%. Figure 4a shows the change in the component concentration at the solidification front depending on the solid phase share (dendrite size) and the cooling rate. The obtained curves illustrate the two essential periods of dendrite growth at the beginning of solidification. The first period, which is characterised by transient conditions of component diffusion in the liquid and occurs with increasing concentration of magnesium at the solidification front, and the second period of growth, which occurs at a relatively equal component concentration at the front and approximately corresponds to the established diffusion conditions. After the liquidus temperature is exceeded, according to the nucleation procedure described in Section 2.3, one *nucleus cell* is placed in the middle of the automaton and the excess component is rejected to cells in Moore’s neighbourhood. These cells become active at temperature *T_N_* where the growth of the dendrite begins. However, due to the very small number of interface cells, the amount of generated solidification heat is still very small and the temperature is further lowered. At the same time, the degree of concentration undercooling and the growth rate of the solid fraction increase. Over time, the dendrite surface (interface) becomes larger and the growth rate is intensified, resulting in the release of more solidification heat and the suppression of the temperature drop. The local minimum for which the solidification heat is equal to the cooling rate is revealed in the solidification curve. The minimum temperature also corresponds to the maximum degree of undercooling of the alloy and the maximum growth rate of the solid fraction.

The further expanding dendrite interface combined with the high growth rate causes the rate of solidification heat generated to exceed the cooling rate, and the metal is heated. The effect of the temperature increase is, in turn, a slight decrease in Mg concentration at the solidification front (Figure 3a and Figure 4a) and the growth rate, which causes the rate of heating the metal to gradually decrease. The system re-establishes a state where the rate of solidification heat is equal to the rate of cooling and a local maximum appears in the solidification curve. The temperature recalescence effect is particularly pronounced at a cooling rate of 45 K/s. With the increase of the cooling rate, the maximum undercooling of the alloy and the temperature range of recalescence increase, and the local minima move to higher shares of the solid fraction (Figure 4b).

### 3.2. Validation of the Numerical Model

In order to determine the potential of the numerical model, the results of the CA simulation were compared with the LGK analytical model. The validation was performed using the guidelines proposed by Zhu and Stefanescu in paper [18]. Figure 5a shows the evolution of the dendrite front growth rate in the model area cooled at the rate of 25 K/s. The values of the growth rate were determined in the interface cells in which the solid fraction was within the range of 0.2–0.8. The presented kinetic pattern indicates that the dendrite growth under transient diffusion conditions takes about 0.2 s. After this time, the growth rate becomes constant and at the same time the highest value, and the further development of the dendrite occurs approximately under the established diffusion conditions. The obtained shape of the growth velocity curve is different from the curves determined in other CA solidification models [15,18,19] and results from the applied nucleation procedure. In the developed model, the growth rate increases from zero to the maximum value, which is physically consistent with the characteristics of the initial solidification period. The achievement of the steady state depends on the rate of heat dissipation from the model area (Figure 5). For the cooling rate of 5 K/s, the time needed to stabilize the dendrite front growth rate is the longest and amounts to more than 1 s. Figure 5 also presents the effect of cooling rate on the value of concentration undercooling for the steady state.

The comparison of the predictive parameters of the LGK model and the developed CA model is presented in Figure 6a–c. The tests were carried out by determining the shape of the dendrite front along with its approximation by the parabolic function for each undercooling value. The applied parabolic approximation ensures a high degree of matching to the values calculated from the model. The determination coefficient *R*^2^ reaches a value above 0.99. Figure 6a shows the calculation results for a dendrite with the preferred crystallographic orientation 0°, growing with undercooling 4.2 K. Based on the obtained approximating functions, the curvature and then the diameter of the dendrite front were determined using dependence *K* = d^2^*y*/d*x*^2^ × [1 + (d*y*/d*x*)^2^]^−3/2^ and the relation *R* = 1/*K*. The comparative characteristics of the dendrite tip diameter calculated from the numerical model and the LGK analytical model for different undercooling are shown in Figure 5b. The LGK model prediction was performed assuming stability parameter σ = 1/4π^2^. When evaluating the test results, it can be observed that both the obtained parameter value range *R* as well as the tendency of its changes depending on undercooling are satisfactory. The second real parameter that allows for validating the numerical model is the dendrite front growth rate under the set diffusion conditions. Simulated growth rate (Figure 6c) also presents a good agreement with the results predicted by the LGK model. However, it should be noted that the values calculated from the numerical model are always below the LGK curves. This fact was found in other papers [18,19,36] and is explained by the sensitivity of the LGK model to parameter σ, which changes depending on undercooling and the initial alloy composition.

### 3.3. Multiple Dendrite Growth Simulation

Simulation of the growth of many dendrites was conducted in the area of 816 × 816 µm^2^ using a cellular automaton with elementary cells sized 408 × 408. The diffusion equation was solved assuming the same boundary conditions as in the previous simulations. On the other hand, for the heat conduction equation on the lower and upper walls, a periodic condition was assumed, and on the right wall—thermal symmetry *q*_b_ = 0, and on the left wall—Newton’s boundary condition:(16)$$q_{b}=\alpha (T-T_{\infty })$$
where: *α*—the heat transfer coefficient, *T_∞_*—the bulk temperature of the cooling media.

The values of the heat transfer coefficient were selected so as to correspond to the DDTA test conditions presented in the next chapter. Calculations were started assuming the initial temperature *T*_0_ equal to 670 °C. After reaching the liquidus temperature, 12 *nucleus cells* with the initial composition *kC*_0_ and the preferred crystallographic orientation ranging from 0° up to 90° were placed randomly in the calculating domain. Coupled motion and interaction of equiaxial dendritic crystal growth during solidification of Al + 5 wt.% Mg alloy is presented in Figure 7.

In the initial period (Figure 7a–c) main branches of dendrites with the preferred crystallographic orientation develop from *nucleus cells*, and their growth is related primarily to the increase in longitudinal dimensions. At the same time, on the surface of the main arms, below the front, seeds of the secondary arms are initiated. The size and shape of the dendrites formed during this period depend on the initial distribution of *nucleus cells*, preferential growth direction and coupled interaction between the crystals. The longest main branches are characteristic of dendrites, the nuclei of which were located at a great distance from others, and the free growth of the arms to the liquid regions lasted the longest. Mutual close location of *nucleus cells* causes the interaction between the main arms to occur in the initial stage of solidification, significantly reducing their size. Inhibition of the longitudinal growth of the main branches causes their thickening and the development of secondary branches. The evolution of the secondary branches is similar to that of the main branches. The possibility of free growth in the directions consistent with the preferred crystallographic orientation contributes to the formation of the longest secondary arms and even to the incubation of the tertiary arms. If, on the other hand, the main branches are parallel to each other and at a short distance, then the secondary limbs disappear and further growth of the dendrite is continued by thickening. At the end of solidification, fusion of the secondary arms occurs, the dendrite interfaces move slightly closer together, and a thin layer of liquid metal remains between them. Computer simulations were completed when the solid fraction in the entire calculating domain reached the value of 0.998.

The developed CA model was compared with the Scheil model and the equilibrium solidification model. The solidification curves obtained from the models are shown in Figure 8. To determine dependency *T^CA^*(*f*_S_), the average temperature from the entire calculating domain was assumed.

Under equilibrium conditions, Al + 5 wt.% Mg alloy solidifies in the temperature range 630–591 °C and has a single-phase structure. The Scheil model considers the alloy solidification in the absence of diffusion in the solid phase, which in turn leads to component segregation and the appearance of a double eutectic in the final solidification period, clearly increasing the solidification temperature range to 630–451 °C. Figure 8 shows that the curve obtained from the CA model lies between the curves specified in the analytical models. This location results from the inclusion of the diffusion equation in the numerical model. The diffusion equation describes the transport of the alloy component in the liquid and solid phases during solidification. The shift of initial curve *T^CA^*(*f*_S_) towards lower temperatures is caused by concentration and capillary undercooling, which determine the dendritic growth rate. Location of the end of solidification temperature (*T_S_^CA^*) in the CA model depends mainly on the rate of diffusion in the solid phase and for the given cooling conditions it occupies an intermediate place (532 °C) between the equilibrium solidus temperature (*T_S_*) and the temperature of the eutectic transformation (*T_E_*).

## 4. Experimental Tests

### 4.1. Tests of the Solidification Process Using the DDTA Method

Solidification is a heterogeneous process and proceeds with different intensity at each point (small fragment) of the casting. The greatest advantage of CA models is the ability to describe the crystal growth under the conditions of microsegregation of the component in the liquid and solid phase, taking into account surface phenomena at the interface. Such a description reflects the actual solidification conditions to a large extent. However, due to the scale of the problem under consideration—the movement of the solidification front in the cells of the automaton with a size of 1 µm and its complex mathematical description, no formulations have been developed at present that would allow for the coupling of microscopic phenomena with the macroscopic course of solidification. The use of an approach analogous to that in micro-macro models is numerically ineffective even when applying multi-processor computing platforms. The purpose of the DDTA method was to compare the actual solidification course in the thermal centre of the casting with the virtual solidification process modelled for this region. The possibility of simulating the crystal growth only on a small fragment of the casting of 1–2 mm is the main limitation of the CA morphological models.

Solidification tests were performed on a synthetic Al + 5 wt.% Mg alloy made of technically pure ingredients: Aluminium—Al 99.9 and Magnesium—Mg 99.9. Melting of the metal charges was carried out in a carborundum crucible in the LEYBOLD HERAEUS IS1/III medium frequency electric induction furnace (LEYBOLD HERAEUS GmbH, Cologne, Germany). The liquid alloy was superheated to a temperature of 670 °C and cast into a metal mould at a temperature of approx. 25 °C. During the casting’s cooling and solidification, the temperature change in a function of time *T(t)* and its derivative *dT/dt* were recorded. DDTA (Derivative differential thermal analysis) tests were performed using the Crystaldigraph module coupled with a PC. The proprietary software DDTA windows version 1.1 was used for data recording and saving. The temperature was measured with a NiCr-NiAl sheathed thermocouple with a diameter of 1.5 mm which was placed in the heat centre of the casting (Figure 9a). The analysed casting had the shape of a plate with dimensions of 12 × 100 × 200 mm. The temperature recording range covered the range from 700 °C up to 500 °C. Measurements were performed three times with time step Δt equal to 0.2 s. Temperature measurement error with DDTA apparatus was 1 °C. Representative curves T(t) and dT/dt revealing solidification thermal effects are shown in Figure 9b.On the basis of the DDTA diagram, the characteristic points of the phase change were read: *T_N_—*the temperature of the onset of solidification, δ*_T_* = *T_3_ − T_1_—*temperature range of recalescence, *T_1_, T_3_—*temperatures at which, respectively, maximum and minimum undercooling of the alloy occur in the initial solidification period *T_2_—*maximum thermal effect of solid phase growth, *T_K_—*temperature of the solidification end.

In order to compare the actual course of cooling and solidification of the Al + 5 wt.% Mg alloy, there is also a curve *T(t)* obtained from numerical simulations in Figure 9b. Both courses are very consistent in terms of shape, location of characteristic phase transition points and cooling and solidification times. The actual onset of solidification and the maximum undercooling temperature of the alloy are 627 °C and 624 °C and are lower, respectively, by 2 °C and by 1 °C in relation to temperatures *T_N_* and *T_1_* calculated from the numerical model. The actual end of solidification temperature is 538 °C and compared with the temperature *T_S_* from the CA model is higher by 6 °C. It should be noted here that the difference between the equilibrium and actual solidus temperature is 55 °C. The total time needed to dissipate the heat of overheating, solidification and cooling down to the temperature of 500 °C in the thermal centre of the casting in real conditions is equal to 40 s. This time, calculated on the basis of the CA solidification model, is slightly shorter by 1.5 s. In order to determine the compliance of the numerical simulation results with the DDTA experimental measurements, the percentage error of the numerical simulation was calculated using the dependence:(17)$$\delta_{S}=\left|\frac{S-R}{R}\right|\cdot 100\%$$
where: *S*—simulation result, *R*—actual result.

Obtained values δ*_S_* (Table 1) indicate that the values of the actual characteristic solidification points are very close to the simulated values. The maximum error of numerical simulations does not exceed 8%, and its average value, taking all the assessed values into account, is 1.92%. Further assessment of the adequacy of the numerical model was performed based on the kinetics of the growth of the solid phase. In the DDTA method, the temperature is measured in the thermal centre of the casting, for which the energy balance is determined by the dependence:(18)$$\frac{dT(t)}{dt}=-\frac{\alpha F(T-T_{{}_{\infty }})}{V\rho c}+\frac{L}{c}\frac{\partial f_{S}}{\partial t}$$
where: *F*—casting surface, *V*—casting volume, *V*/*F*—reduced casting wall thickness.

Above the liquidus temperature and below the solidus temperature, the integration of Equation (18) gives a solution in the form:(19)$$\frac{T(t)-T_{\infty }}{T_{Z}-T_{\infty }}=\exp\left(-\frac{\alpha F}{V\rho c}t\right)=\exp\left(-\xi t\right)$$
which allows for calculating the heat transfer coefficient—*α* and the cooling rate of the casting—*ξ*, with a known die pouring temperature—*T_Z_* and characteristics *T*(*t*). The values of these parameters determined on the basis of DDTA curves (Figure 9b) are: α*_L_ =* 336 Wm^−2^K^−1^*, ξ_L_ =* 19.1 × 10^−3^
*s*^−1^ for *T* > *T_L_*, and α*_S_ =* 268 Wm^−2^K^−1^*, ξ_S_ =* 15.6 × 10^−3^*s*^−1^ for *T* < *T_S_*, In the solidification temperatures range *T_L_**–**T_S_* their change is assumed to be linear. Based on the thermal parameters of the metal mould, the “base curve” (red line in Figure 9b), which shows the change in the casting cooling rate in the absence of internal heat sources, was determined. The area limited by the blue and red curves (Figure 9b) shows the thermal effect related to the heat of solidification, and integration of this area determines the kinetics of the solid phase growth (Figure 10).

Based on the obtained courses of *f_S_*(*t*), it can be concluded that, in terms of solidification kinetics, the results of computer simulations also correspond to the experimental results of DDTA. The difference between the calculated solidification time and the actual solidification time is 2.5 s, and the average growth rate of the solid fraction (determined from the slope of the curves *f_S_*(*t*) in the middle solidification period (0.2–0.8) *f_S_*) is equal to 0.035 s^−1^ for CA model and 0.039 s^−1^ for DDTA tests.

### 4.2. Results of Microstructural Tests

Microstructural tests of Al + 5 wt.% Mg alloy was performed using a Phenom XL scanning microscope equipped with an EDS analyser (PhenomWorld, Eindhoven, The Netherlands). The test samples were taken from the casting area where the NiCr-NiAl thermocouple was installed. The typical alloy microstructure and the magnesium concentration profile determined on the section oriented perpendicular to the secondary arms of the dendrite are presented in Figure 11a,b. Figure 11c,d show the virtual structure of the alloy and the Mg concentration profile calculated by the CA method, also determined on the section oriented perpendicular to the secondary arms of the dendrite. The mean SDAS value obtained from numerical calculations is 48μm, while the experimental value of this parameter is 51 μm. Based on the research on the chemical composition, it can be concluded that the minimum concentration of Mg in the areas of the secondary arms axes is at the level of 3.4% and 3.1% for the CA model and actual chemical analyses, respectively. The maximum Mg concentration at the interface of the secondary arms reaches the value of about 9.5% in numerical simulations and about 11.5% in the EDS analysis. The differences in the maximum values, apart from the possible inaccuracy of the numerical model, may be caused by local Mg microsegregation in the final solidification phase or by precipitation processes occurring in the solid phase, which result from the decreasing solubility of Mg in the solution α(Al) as the temperature drops. Nevertheless, it can be concluded that, in terms of quantity and quality, the results of numerical simulations demonstrate a high agreement with the experimental studies of the microstructure.

## 5. Conclusions

A 2D model was developed to simulate equiaxial dendritic growth in a binary alloy. The model uses the methodology for calculating the solid phase increments in the interface cells, which takes the change in the concentration of the component in the solid and liquid phase during the solidification transition into account. The procedure of rejecting the component to the cells from the nearest neighbourhood used in the calculations allows for mapping the dynamics of dendrite growth from initially unsteady diffusion conditions to a steady state. The developed model allows for a very realistic reproduction of dendrite features, such as: the parabolic shape of the dendrite front, incubation of the nuclei along the main arms of the dendrite, the evolution of the secondary arms, and the coupled influence of the movement of local solidification fronts and inhibition of the growth of equilateral dendrites. In the tested Al + 5 wt.% Mg alloy, the shape and size of the primary and secondary arms depend mainly on the cooling rate. At low cooling rates, only main dendrite branches form, while at rates above 25 K/s, secondary branches develop. Dendritic growth under initially unsteady diffusion conditions induces temperature recalescence, the extent of which increases with the increasing cooling rate. The final shape of the generated dendritic structure depends on the initial distribution of *nucleus cells* in the calculating domain, the preferred growth directions of the dendrite arms, their mutual orientation and interaction during solidification. In order to determine the potential of the numerical model, the simulation results were compared with predictions of the LGK analytical model. It was demonstrated that the variability of the dendrite tip diameter and the growth rate determined in the CA model are similar to the values obtained in the LGK model. The correctness of the numerical model of CA solidification was confirmed by experimental tests of DDTA and EDS. The obtained solidification curves of the Al + 5 wt.% Mg alloy, the mean value of SDAS, as well as the profiles of magnesium concentration, demonstrated a high agreement with the results of the calculations, both in terms of the range of values of individual quantities and their change tendencies. The mean value of the solidification simulation error was 2%. It should be noted, however, that the developed CA model allows for the simulation and analysis of the solidification process only of small fragments of the casting. The comparative analysis of the test results shows that the developed CA model, in quantitative and qualitative terms, adequately describes the actual solidification process and allows for replicating the realistic microstructure of the Al + 5 wt.% Mg alloy.

## Figures and Tables

**Figure 1 materials-14-03393-f001:**
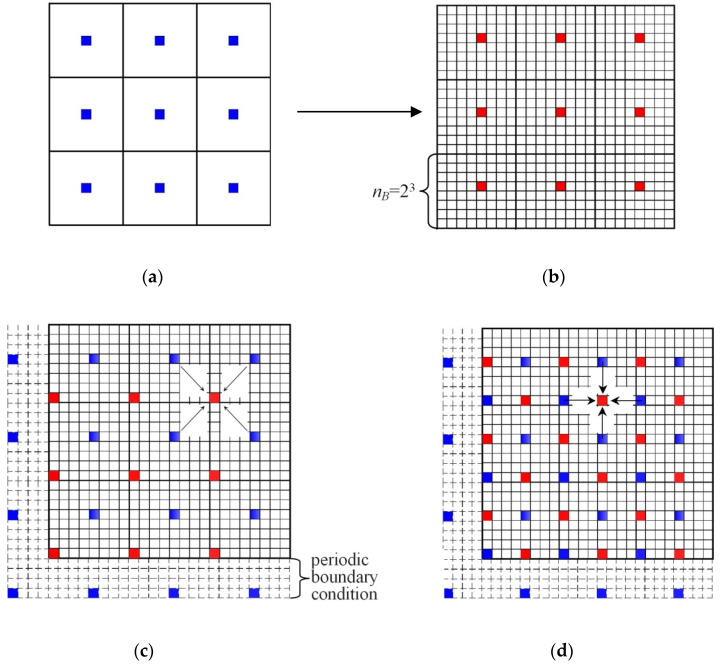
Diagram of temperature interpolation on a fragment of the main grid with blocks of size 2^3^ × 2^3^ for two consecutive steps of calculations. (**a**) calculation of temperature field on coarse lattice; (**b**) Transfer of temperatures from a coarse lattice to central cells of CA blocks; (**c**) step 1. calculation of average temperature on the basis of neighborhood $\left(\pm 2^{n-1}a\sqrt{2^{}},\pm 2^{n-1}a\sqrt{2^{}}\right)$; (**d**) step 2. calculation of average temperature on the basis of neighborhood $\left(\pm 2^{n-1}a,\pm 2^{n-1}a\right)$.

**Figure 2 materials-14-03393-f002:**
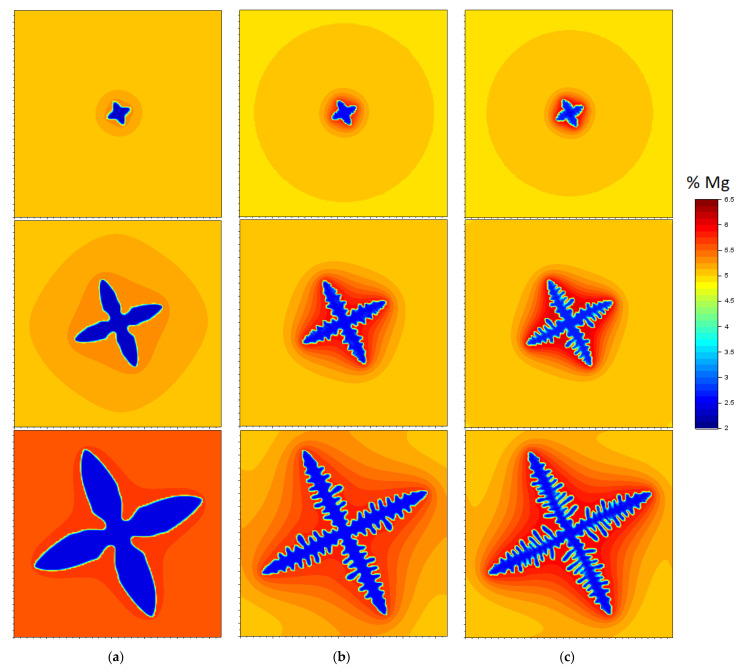
Instantaneous Mg concentration fields and free dendrite growth sequences in the Al + 5 wt.% Mg alloy cooled at the rate: (**a**) 5 K/s, (**b**) 25 K/s, (**c**) 45 K/s.

**Figure 3 materials-14-03393-f003:**
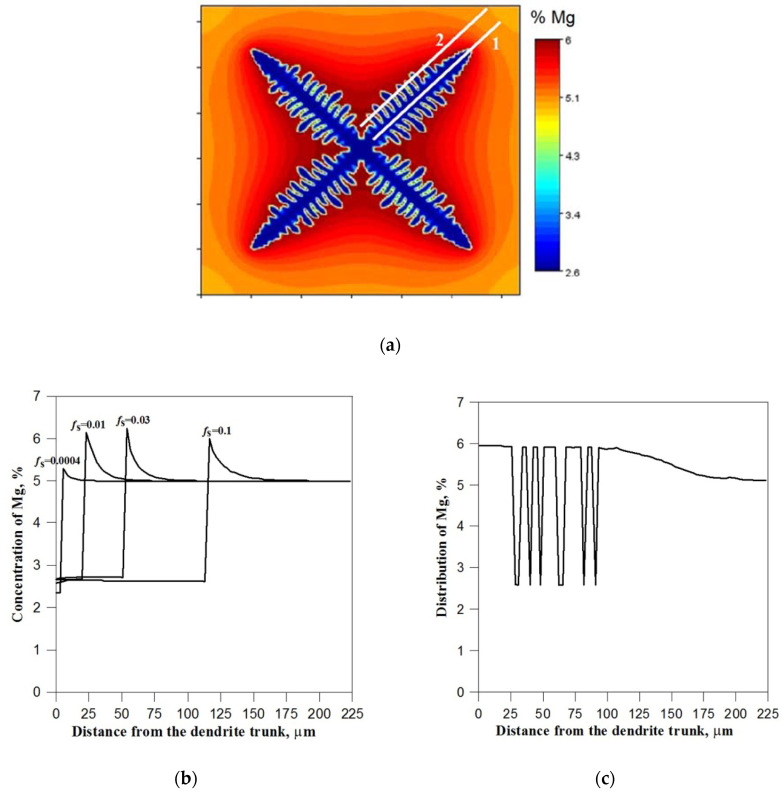
The results of the simulation of the equiaxial dendrite growth for the cooling rate of 45 K/s: (**a**) the morphology of the dendrite with orientation *θ*_0_ = 45°, (**b**) Mg concentration profiles during dendrite growth determined along line 1, (**c**) magnesium distribution along the section starting from the base of the dendrite and crossing the secondary arms (line 2) − *f*_S_ = 0.1.

**Figure 4 materials-14-03393-f004:**
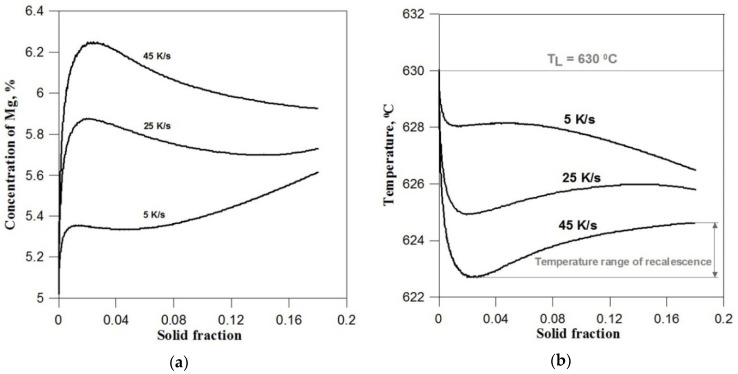
(**a**) changes in the concentration of magnesium at the solidification front depending on the share of the solid phase (dendrite size) and different cooling rates, (**b**) changes in the average temperature in the initial period of crystal growth.

**Figure 5 materials-14-03393-f005:**
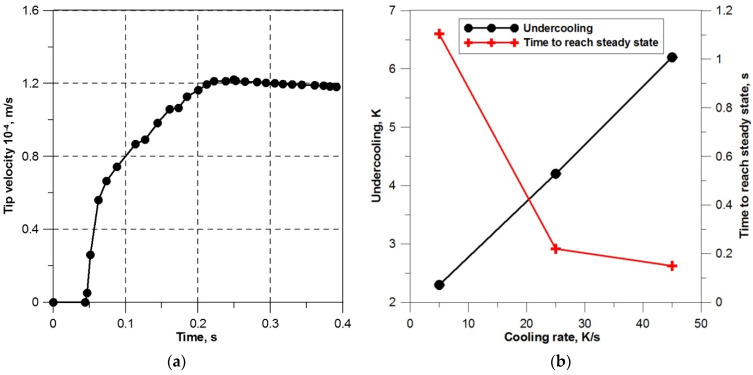
Kinetic characteristics of the initial solidification period: (**a**) the growth rate of the dendrite front, (**b**) the effect of the cooling rate on the concentration undercooling value and the transition time from the transient state to the steady state.

**Figure 6 materials-14-03393-f006:**
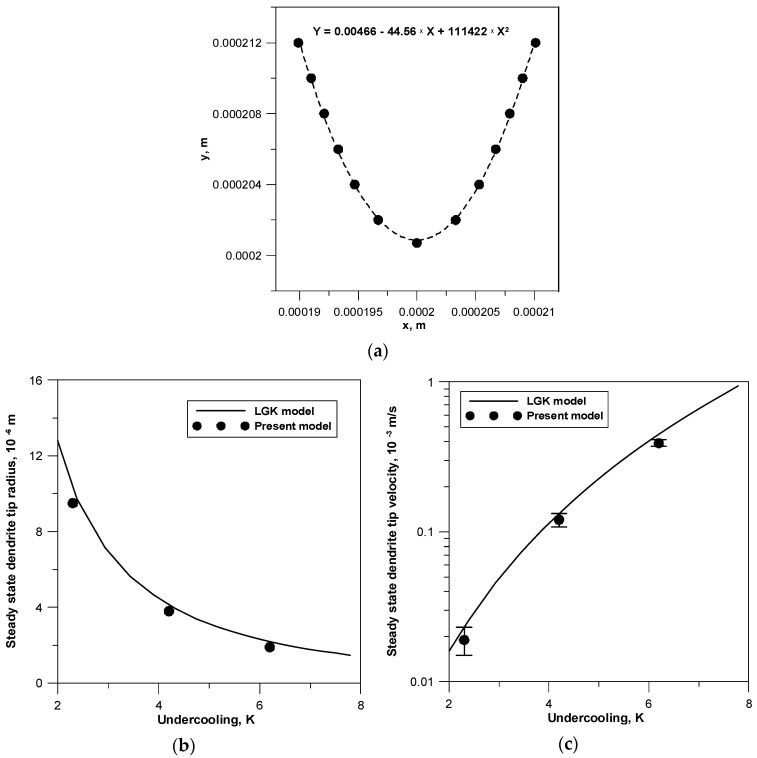
Comparison of the predictive parameters of the LGK model and the developed CA model: (**a**) the shape of the dendrite front and its parabolic approximation, (**b**) the effect of concentration undercooling on the dendrite front diameter, (**c**) the effect of concentration undercooling on the velocity of the dendrite front.

**Figure 7 materials-14-03393-f007:**
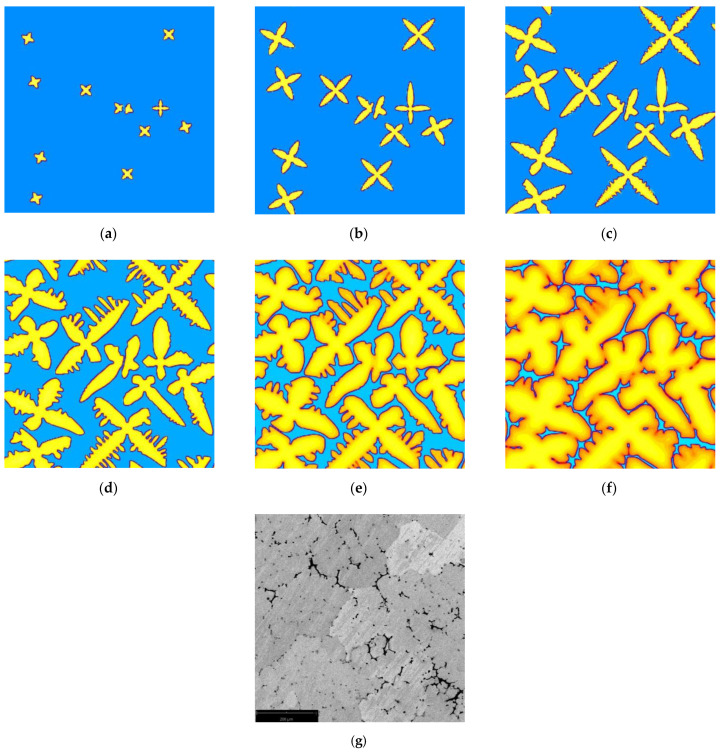
Multi-dendritic growth sequence in an solidifying Al + 5 wt.% Mg alloy: (**a**) 0.02, (**b**) 0.1, (**c**) 0.2, (**d**) 0.45, (**e**) 0.7, (**f**) 0.9 of solid fraction, (**g**) actual microstructure of the Al + 5 wt.% Mg alloy cast into a metal mold (Section 4).

**Figure 8 materials-14-03393-f008:**
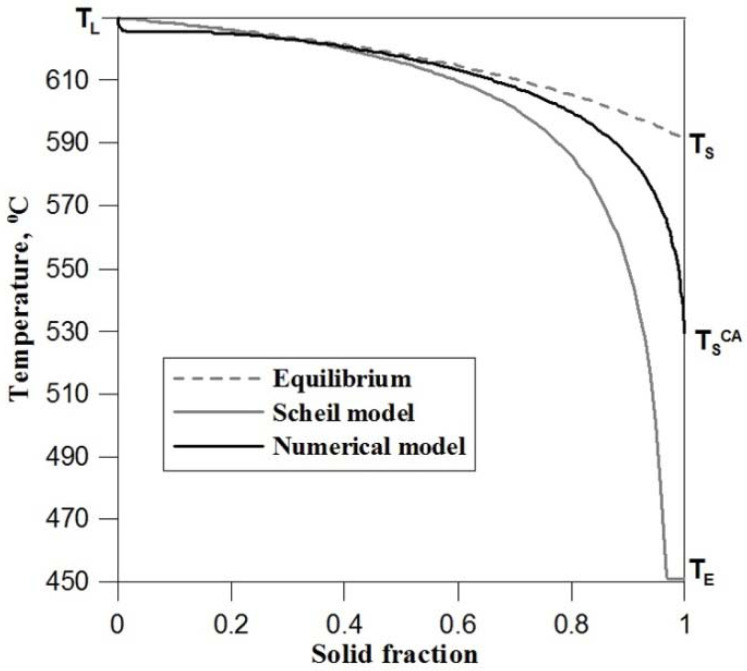
Solidification curves determined from the numerical CA model, equilibrium model and Scheil model.

**Figure 9 materials-14-03393-f009:**
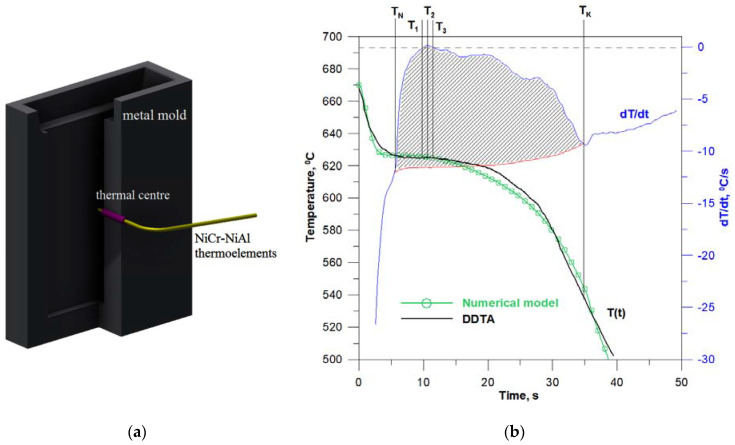
DDTA test results: (**a**) appearance of a metal mold with a thermocouple, (**b**) solidification and cooling curves of the Al + 5 wt.% Mg alloy.

**Figure 10 materials-14-03393-f010:**
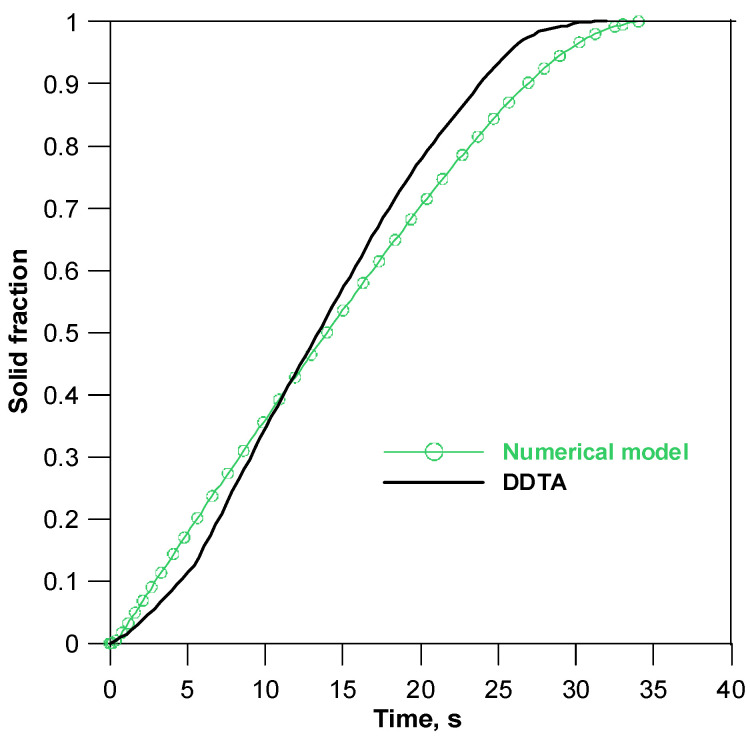
Comparison of the growth kinetics of the solid fraction calculated in the CA solidification model with DDTA tests.

**Figure 11 materials-14-03393-f011:**
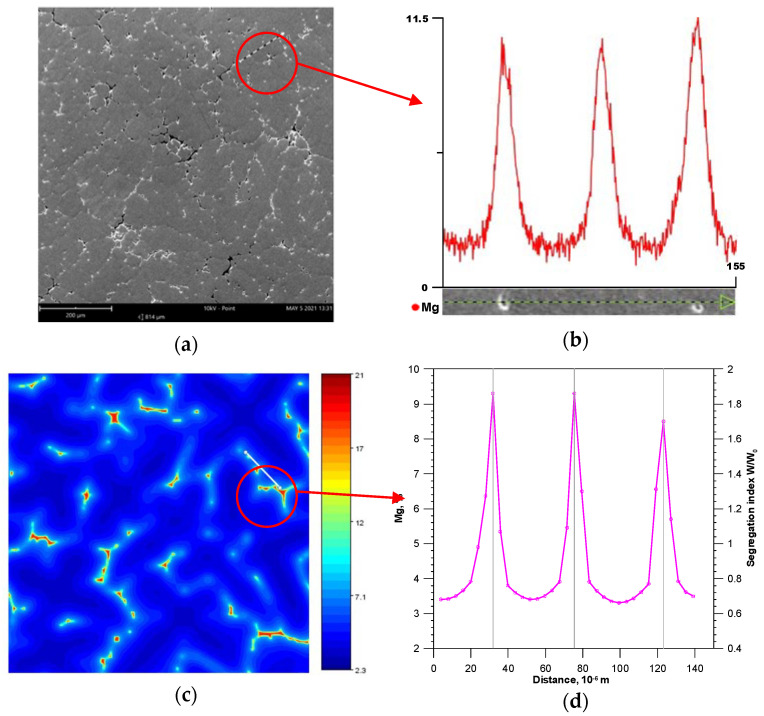
Microstructure test results: (**a**) die casting (**b**) magnesium concentration profile in the selected area of the die casting, (**c**) surface Mg distribution calculated from the CA model (*f*_S_ = 0.99) (**d**) magnesium concentration profile determined from the CA model.

**Table 1 materials-14-03393-t001:** Comparison of DDTA test results with results of solidification simulation (CA).

Characteristic Quantity	Marking	DDTA Tests	The Numerical Model	Simulation Error, %
The temperature of the onset of solidification, °C.	*T_N_*	627	629	0.32
Temperature of maximum undercooling of the alloy, °C.	*T* _1_	624	625	0.16
Temperature range of recalescence, °C.	δ*_T_ = T*_3_ *– T*_1_	1	1	0.00
Maximum thermal effect of solid phase growth, °C.	*T* _2_	625	626	0.16
The end of solidification temperature, °C.	*T_K_*	538	532	1.12
Solidification time, s	*t_K_ – t_P_*	31.5	34	7.94
The total time of solidification and cooling to the temperature of 500 °C, s	*t* _500_ *–t* _0_	40	38.5	3.76

## Data Availability

Data sharing is not applicable to this article.

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
