# Peer review of "CA Modeling of Microsegregation and Growth of Equiaxed Dendrites in the Binary Al-Mg Alloy"

_materials, 2021, doi:10.3390/ma14123393_

Round 1

Reviewer 1 Report

This paper constructed a two-dimensional model based on the cellular automation for simulating free dendritic growth in the Al-Mg alloy. The development of microsegregation in the liquid and solid phase were analyzed during solidification. The comparative tests were carried out between numerical simulations and actual solidification, showing a high agreement. The content of the paper is substantial. But there are still some problems in this manuscript:

  1. In the section 2.2, a sparse grid was used to improve the computational efficiency. How to determine the size ratio between sparse grid size with automation cell size to ensure the accuracy of interpolated temperature?
  2. In the line 447, there should be Figure 5.
  3. Please unify legends on range and add units.
  4. The dentrite growth simulation is an important part in this paper. So, it is recommended to add some solidification structure or morphology analysis to validate the simulation results.
  5. In the section 4.1, the number of conducted experiments is missing, and errors/deviations of mold centre temperatures are lacking in Figure 9.
  6. In the Figure 11 c), the highest magnesium concentration is up to 21% while the curve in d) shows not. Please justify your choice of lines in a) and c) (in terms of comparativeness).

Reviewer 2 Report

The aim of the paper is to present a 2D model to simulate equiaxial dendritic growth in a binary alloy. The study of new models for numerical simulations for dendritic growth in alloys for casting applications is nowadays an interesting subject from the scientific and industrial point of view however, unfortunately, the research developed by the authors presents certain limitations.

1.- The author uses a combination of the cellular automaton technique and the control volume method performing the calculations on a flat area divided by regular square grids and elementary cells of side "a" for the modeling of the dendritic structure evolution. Later, it validates the numerical results of its simulation with an experimental analysis on a casting part whose geometry is a regular plate with dimensions of 12 * 100 * 200 mm. The experimental data of temperature were taken with a thermocouple located in the center of the casting mold.

The casting part is usually characterized by presenting a complex geometry of large dimensions and small details, changes in thickness, etc. These complex geometric features can influence the experimental thermal results compared to a standard flat part. The author is recommended to detail the limits of his research in terms of its application to real cases of parts to be manufactured with the casting process since the case study included presents a geometry of great simplicity.

2.- The author presents a constant precision 2D model based on regular square grids and elementary cells of side "a". In the numerical simulation of complex models, the precision of the algorithm is a crucial factor in obtaining results consistent with reality. The author is recommended to detail the influence of the value “a” of the cells on the precision of the presented algorithm. Likewise, the use of a 2D model can also present problems in its application to complex geometries. In these cases, the use of voxelized models of variable precision can be useful. The author is recommended to detail the limits of his research in terms of the precision of his algorithm and the use of the 2D model in parts of complex geometry.

Round 2

Reviewer 2 Report

The authors have addressed satisfactorily the points raised during the review.Therefore, I recommend the publication of this article.